# Cranial Neural Crest Specific Deletion of *Alpl* (TNAP) via P0-Cre Causes Abnormal Chondrocyte Maturation and Deficient Cranial Base Growth

**DOI:** 10.3390/ijms242015401

**Published:** 2023-10-20

**Authors:** Naoto Ohkura, Hwa Kyung Nam, Fei Liu, Nan Hatch

**Affiliations:** 1Department of Orthodontics and Pediatric Dentistry, School of Dentistry, University of Michigan, Ann Arbor, MI 48109, USA; ohkura@dent.niigata-u.ac.jp (N.O.); hknam@umich.edu (H.K.N.); 2Division of Cariology, Operative Dentistry and Endodontics, Department of Oral Health Science, Niigata University Graduate School of Medical and Dental Sciences, Niigata 951-8510, Japan; 3Department of Biomaterials Sciences and Prosthodontics, School of Dentistry, University of Michigan, Ann Arbor, MI 48109, USA; feiliu@umich.edu

**Keywords:** tissue nonspecific alkaline phosphatase, hypophosphatasia, cranial neural crest, cranial base, synchondrosis, chondrocyte, phosphate, pyrophosphate

## Abstract

Bone growth plate abnormalities and skull shape defects are seen in hypophosphatasia, a heritable disorder in humans that occurs due to the deficiency of tissue nonspecific alkaline phosphatase (TNAP, *Alpl*) enzyme activity. The abnormal development of the cranial base growth plates (synchondroses) and abnormal skull shapes have also been demonstrated in global *Alpl*^−/−^ mice. To distinguish local vs. systemic effects of TNAP on skull development, we utilized P0-Cre to knockout *Alpl* only in cranial neural crest-derived tissues using *Alpl* flox mice. Here, we show that *Alpl* deficiency using P0-Cre in cranial neural crest leads to skull shape defects and the deficient growth of the intersphenoid synchondrosis (ISS). ISS chondrocyte abnormalities included increased proliferation in resting and proliferative zones with decreased apoptosis in hypertrophic zones. ColX expression was increased, which is indicative of premature differentiation in the absence of *Alpl*. Sox9 expression was increased in both the resting and prehypertrophic zones of mutant mice. The expression of Parathyroid hormone related protein (PTHrP) and Indian hedgehog homolog (IHH) were also increased. Finally, cranial base organ culture revealed that inorganic phosphate (P_i_) and pyrophosphate (PP_i_) have specific effects on cell signaling and phenotype changes in the ISS. Together, these results demonstrate that the TNAP expression downstream of *Alpl* in growth plate chondrocytes is essential for normal development, and that the mechanism likely involves Sox9, PTHrP, IHH and PP_i_.

## 1. Introduction

Long bone growth occurs at specific locations known as growth plates, where chondrocytes undergo a well-regulated process of proliferation and differentiation to form cartilage, which is then replaced by bone via apoptosis of terminally differentiated hypertrophic chondrocytes, and invasion of blood vessels to provide bone cells for bone formation [1,2]. This endochondral bone growth process also occurs in the skull bones of the cranial base, upon which the brain sits. In contrast to long bones, the cartilagenous bones of the cranial base are bidirectional, with a central resting zone surrounded on both anterior and posterior sides by proliferating, prehypertrophic and hypertrophic zones [3,4]. Cranial base growth plates are known as synchondroses. The two primary synchondroses of the cranial base in mice are derived from different embryonic origins. The spheno-occipital synchondrosis (SOS) is initially derived from a mixed neural crest and paraxial mesoderm origin, while the intersphenoid synchondrosis (ISS) is of cranial neural crest origin [5]. During postnatal development, the mesoderm/neural crest boundary within the SOS migrates rostrally such that the SOS becomes solely of mesoderm origin by several days after birth [5,6]. The abnormal development and premature fusion of cranial base synchondroses is associated with numerous genetic disorders, leading to the deficient anterior–posterior growth of the skull, associated skull shape defects and malocclusion (teeth not fitting together appropriately).

Hypophosphatasia is a rare inherited disorder caused by inactivating mutations in the gene (Alpl) that encode the enzyme tissue non-specific alkaline phosphatase (TNAP). One primary and well-established function of TNAP is to hydrolyze inorganic pyrophosphate (PP_i_) into inorganic phosphate (P_i_) [7]. Because PP_i_ inhibits matrix mineralization while P_i_ serves as a substrate for hydroxyapatite formation, TNAP activity promotes bone mineralization [8]. Both humans and mice that are affected by TNAP deficiency exhibit skeletal hypomineralization [9,10,11]. The influence of TNAP on skeletal development is not limited to mineralization. Humans and mice affected by TNAP deficiency also exhibit abnormal growth plates [10,12].

We previously investigated skull phenotype changes in the global *Alpl*^−/−^ mouse model of hypophosphatasia to better understand mechanisms leading to craniosynostosis (the premature fusion of calvarial/skull cap bones, leading to high intracranial pressure) in a disorder of bone hypomineralization. We found evidence of skull bone hypomineralization, craniosynostosis and an abnormal skulls shape, which are also seen in infantile hypophosphatasia [11,13]. We also found evidence of the abnormal development of cranial base synchondroses in these mice [14]. TNAP is highly expressed in hypertrophic zone chondrocytes [10], and we observed the expansion of the hypertrophic zone in global *Alpl*^−/−^ mice. The isolation of rib chondrocytes from *Alpl*^−/−^ mice revealed a cell autonomous influence of *Alpl* deficiency on chondrocytes [15].

Overall, our analyses of global *Alpl*^−/−^ mice indicated that *Alpl* influenced the skull bones of the cranial neural crest more than those of paraxial mesoderm origin [14,15]. In addition, global *Alpl*^−/−^ mice are phenotypically normal at birth but rapidly decline in health, with most dying before or shortly after weaning. Therefore, here we investigated the impact of *Alpl* deletion specifically in tissues of cranial neural crest origin at the early and later stages of postnatal development by crossing *P0-Cre* promoter expressing mice with floxed *Alpl* mice for cranial neural crest expression [16].

Results demonstrate that the deletion of *Alpl* in cranial neural crest via P0-Cre does not cause craniosynostosis; however, it does cause the abnormal development of the anterior (neural crest origin) but not posterior (mesoderm origin) cranial base, as well as an abnormal skull shape. *Alpl*^fl/fl^; *P0-Cre^+^* mice exhibit abnormal chondrocyte proliferation, maturation and apoptosis, as well as aberrant Sox9, PTHrP and IHH signaling in the intersphenoid synchondrosis ISS, as compared to *Alpl*^fl/fl^ mice. The organ culture of the wild type cranial base showed that P_i_ and PP_i_ regulate PTHrP and IHH signaling as well as chondrocyte maturation. Because P_i_ and PP_i_ levels are anticipated to be down and up, respectively, upon *Alpl* deletion, these latter data indicate that the chondrocyte changes in vivo may be mediated by locally increased PP_i_. Additionally, the knockdown of *Alpl* by shRNA in the O9.1 neural crest cell line showed abnormalities in the markers of differentiation and stemness, affirming a cell-autonomous role for TNAP in neural crest cells.

## 2. Results

### 2.1. Alpl Deletion by P0-Cre Does Not Cause Craniosynostosis but Does Lead to Skull Shape Abnormalities

*Alpl*^fl/fl^; *P0-Cre^+^* mice are born healthy and are similar in size and weight compared to control mice. By 35 days after birth, *Alpl*^fl/fl^; *P0-Cre^+^* mice remain healthy and viable but are diminished in size and weight compared to control mice (Figure 1A). An abnormal skull shape including midface deficiency is apparent in *Alpl^fl/fl^*; *P0-Cre^+^* mice by day 35 (Figure 1B). Skull linear measurements that are normalized for overall skull size reveal an increased cranial height; decreased inner canthal distance; and decreased nasal bone, nose, frontal bone and overall skull lengths (Figure 1C). Skull width and parietal bone length were not different between *Alpl^fl/fl^; P0-Cre^+^* and *Alpl*^fl/fl^ mice. It is important to state that, although the individual linear distances between the genotypes appear small on the graph, this is because the measurements are presented as normalized to total skull length (nasale to opisthion). It is also important to note that these individual skull linear measurement differences add up to a grossly abnormal skull shape, as seen in Figure 1B. Finally, no evidence of any type of craniosynostosis was found in any of the mice, regardless of genotype.

### 2.2. P0-Cre Alpl Deletion Causes Abnormal Development of the Anterior but Not the Posterior Cranial Base

Nano-CT images revealed obvious abnormalities including fusions of the ISS but not the SOS (Figure 2A). Eighty percent (80%) of *Alpl*^fl/fl^; *P0-Cre^+^* mice had partial or complete fusion of the ISS, while no *Alpl*^fl/fl^ mice had no evidence of fusion (*p* < 0.00001, Fisher exact test). The raw linear measurements of cranial base bones revealed an 11.4% decrease (*p* < 0.05) in presphenoid bone length, a 9.6% decrease (*p* < 0.05) in basisphenoid bone length, with no change in basioccipital bone length in *Alpl*^fl/fl^; *P0-Cre^+^* as compared to *Alpl*^fl/fl^ mice. On the other hand, when each bone length was normalized for skull length (because the *Alpl* conditional knockout mice had smaller skulls than the control mice), there was a 10.0% decrease (*p* < 0.05) in presphenoid bone length, a 7.7% decrease (*p* < 0.05) in basisphenoid bone length and a 4.0% increase (*p* < 0.05) in basioccipital bone length in *Alpl*^fl/fl^; *P0-Cre^+^* as compared to *Alpl*^fl/fl^ mice (Figure 2B).

To confirm the specificity of *Alpl* knockdown in cranial neural crest-derived tissues (e.g., the ISS and not the SOS), and to assess phenotypic changes within the *Alpl* deficient ISS, we performed the histomorphometric analysis of stained sections from day 5 mice. Alkaline phosphatase staining for TNAP activity was strong in the hypertrophic and prehypertropic zones of both the ISS and SOS of control *Alpl*^fl/fl^ mice. In contrast, alkaline phosphatase staining was strong in the hypertrophic and prehypertropic zones in the SOS but not the ISS of *Alpl*^fl/fl^; *P0-Cre^+^* mice (Figure 3A). Total ISS area was increased in *Alpl*^fl/fl^; *P0-Cre^+^* compared to *Alpl*^fl/fl^ mice (Figure 3B). ISS chondrocyte zone widths, as normalized for total ISS width, demonstrated a significantly increased width of the hypertrophic zone and a significantly decreased width of all other zones in in *Alpl*^fl/fl^; *P0-Cre^+^* compared to *Alpl*^fl/fl^ mice. While there appeared to be a greater area of Alcian Blue staining for the cartilage matrix, there was no difference in staining per cell in any zones of the ISS in conditional *Alpl* knockout, as compared to control mice (Figure 3C). No changes were found in SOS area, chondrocyte zone widths or Alcian Blue staining between *Alpl*^fl/fl^; *P0-Cre^+^* and *Alpl*^fl/fl^ mice (Appendix A).

### 2.3. P0-Cre Alpl Deletion Increases Chondrocyte Proliferation and Decreases Apoptosis in the Anterior Cranial Base Synchondrosis (ISS)

Next, we performed Ki67 staining to quantify proliferating cells in the synchondroses of 5-day-old mice. Ki67 positive staining was mainly present in the resting zone (RZ) and proliferation zone (PZ) of the mice, regardless of genotype (Figure 4A). When comparing genotypes, Ki67 positive staining was significantly greater in the RZ and PZ of *Alpl*^fl/fl^; *P0-Cre^+^* mice compared to *Alpl*^fl/fl^ ISS.

We next sought to determine whether the *Alpl*^fl/fl^; *P0-Cre^+^* mice exhibited diminished synchondrosis chondrocyte apoptosis, as we had previously seen in global *Alpl*^−/−^ mice [15]. While the detection of apoptosis via either TUNEL or Annexin V staining (Annexin V staining not shown) revealed overall low levels of apoptosis in the ISS of 5-day-old mice, quantification revealed significantly decreased levels of apoptosis in both the posterior and anterior hypertrophic zones (HZ) in *Alpl*^fl/fl^; *P0-Cre^+^* as compared to *Alpl*^fl/fl^ mice (Figure 4B).

### 2.4. P0-Cre Alpl Deletion Increased PTH-Related Protein (PTHrP) and Indian Hedgehog (IHH) Expression in the Anterior Cranial Base Synchondrosis (ISS)

Because interactive PTHrP-IHH signaling is known to coordinate diverse aspects of bone morphogenesis in long bone growth plates and cranial base synchondroses [17,18], we hypothesized that aberrant PTHrP-IHH signaling may underly the ISS abnormalities seen in *Alpl*^fl/fl^; *P0-Cre^+^* mice. Immunofluorescent staining for PTHrP in the ISS of 5-day-old mice showed increased PTHrP expression in the resting and hypertrophic zones of *Alpl*^fl/fl^; *P0-Cre^+^* as compared to *Alpl*^fl/fl^ mice (Figure 5A). Immunofluorescent staining for IHH in the ISS of 5-day-old mice showed increased IHH expression in the prehypertrophic and hypertrophic zones of *Alpl*^fl/fl^; *P0-Cre^+^* as compared to *Alpl*^fl/fl^ mice (Figure 5B). No differences were found between genotypes for the SOS (Appendix A).

### 2.5. P0-Cre Alpl Deletion Promotes Premature Chondrocyte Differentiation in the Anterior Cranial Base Synchondrosis (ISS)

To determine whether *Alpl* deletion by *P0-Cre* influences the maturation of chondrocytes in the cranial base synchondroses, we next performed immunofluorescent staining for the sex-determining region Y-box 9 (Sox9), collagen II (Col II) and collagen X (Col X). Sox9 expression was higher in the resting and prehypertrophic zones of *Alpl*^fl/fl^; *P0-Cre^+^* as compared to *Alpl*^fl/fl^ mice (Figure 5C). Col X expression was also significantly higher in *Alpl*^fl/fl^; *P0-Cre^+^* as compared to *Alpl*^fl/fl^ mice in these same chondrocyte zones (Figure 5D). No differences were found between genotypes for Col II expression in the ISS (Appendix A). No differences were found between genotypes for Sox9 or Col X in the SOS (Appendix A).

### 2.6. PP_i_ and P_i_ Differentially Regulate PTHrP-IHH Signaling and Chondrocyte Maturation in Cultured Anterior Cranial Base

A primary substrate for TNAP is PP_i_, which is hyrolyzed to P_i_. Therefore, TNAP deficiency leads to high PP_i_ and potentially low P_i_ levels, which prior studies have shown can influence gene expression and proliferation [19,20,21], in addition to influencing other cellular phenotypes, including chondrocyte differentiation and apoptosis [22]. To determine whether local changes in levels of PP*_i_* and/or P*_i_* could account for the changes seen in the ISS of *Alpl*^fl/fl^; *P0-Cre^+^* mice, we performed the organ culture of newborn anterior cranial base isolated from wild type mice in P_i_ free media supplemented with/without PP_i_ or P_i_. Tissues were stained with H&E or with a colorimetric substrate for TNAP to assess overall phenotype changes (Figure 6A). It was first noted that chondrocyte zones in the anterior aspect of the ISS were difficult to distinguish, so quantifications were only performed in the posterior aspect of the ISS. In addition, due to the difficulty in the ability to clearly distinguish all chondrocyte zones of the posterior ISS, histomorphometry was performed on combined resting and proliferating zones and on the hypertrophic zone. The quantification of stained sections demonstrated that culture with PP_i_ or P_i_ for five days increased ISS area. Culture with P_i_ but not PP_i_ increased the combined width of the resting and proliferative zones. Culture with either P*_i_* or PP*_i_* increased the width of the hypertrophic zone, although PP_i_ did so significantly more than P_i_.

We next performed immunofluorescence staining to quantify levels of IHH and PTHrP (Figure 6B). Results show that P*_i_* but not PP*_i_* significantly increased IHH expression throughout the ISS and specifically as quantified in the combined resting and proliferative zones, and in the posterior hypertrophic zone. In contrast, culture with either P*_i_* and PP*_i_* increased PTHrP expression throughout the ISS and specifically as quantified in the combined resting and proliferative zones and in the posterior hypertrophic zone, although PP*_i_* increased to a significantly greater extent than P*_i_*.

We quantified chondrocyte differentiation and apoptosis markers via immunofluorescence that were shown to be significantly altered in the ISS of *Alpl*^fl/fl^;*P0-Cre^+^* mice (Figure 6C). Staining and quantification revealed that P*_i_* increased while PP*_i_* decreased expression of Col X in the combined resting and proliferative zones and that P*_i_* increased while PP*_i_* had no effect on the expression of Col X in the posterior hypertrophic zones. Similar results were found for Sox9 expression, except that P*_i_* increased while PP*_i_* decreased the expression of Sox9 in the combined resting and proliferative zones and in the posterior hypertrophic zone. TUNEL staining for apoptosis demonstrated that PP*_i_* decreased while P*_i_* had no impact on apoptosis levels in the combined resting and proliferative zones or the posterior hypertrophic zone.

### 2.7. Alpl Deficient O9-1 Cells Influences the Stemness and Differentiation

In the musculoskeletal field, TNAP is most widely known for its role in skeletal growth and mineralization. Yet, TNAP is also expressed in a wide variety of progenitor cells [23,24], including embryonic stem cells [25,26] and induced pluripotent stem cells [27]. To initiate an investigation of a potential TNAP influence on pluripotent neural crest cells, we knocked down the expression of TNAP by stable transfection with *Alpl* (TNAP) shRNA in the O9.1 mouse neural crest cell line [28]. Transfection with *Alpl* but not with nontarget control shRNA significantly decreased alkaline phosphatase activity and expression of *Alpl* mRNA (Figure 7A). To determine whether *Alpl* deletion influences chondrocyte differentiation, the cells were cultured in chondrogenic media then stained with Alcian Blue for cartilaginous matrix accumulation. O9.1 cells expressing *Alpl* shRNA exhibited significantly decreased staining for Alcian Blue when compared to cell expressing nontarget shRNA (Figure 7B). The real-time PCR of mRNA changes revealed that O9.1 cells expressing *Alpl* shRNA expressed similar levels of the neural crest markers *AP-2a*, *Twist1* and *Snail1* but expressed significantly lower levels of Sox9 mRNA as compared to the control cells (Figure 7C). The knockdown of *Alpl* in O9.1 cells decreased the expression of the stem cell markers *Nestin* and *Sca-1* but increased the expression of *CD44*.

## 3. Discussion

The original goal of this study was to determine whether the deletion of *Alpl* in cranial neural crest cells would cause coronal craniosynostosis in mice. This idea was based upon our prior findings, which showed that the skull bones of cranial neural crest origin (e.g., facial bones, frontal bones) appeared to be more affected than those of mesoderm origin (e.g., parietal bones, occipital bones) and that coronal craniosynostosis occurred in approximately 1/3 of global *Alpl*^−/−^ mice with a mixed C57BL/6J-129/SvJ genetic background [14]. This level of incidence of craniosynostosis is similar to that reported to occur in infants and children with hypophosphatasia [13,29,30]. The coronal suture sits at a boundary between mesoderm-derived parietal bones and neural crest-derived frontal bones [31]. Early studies indicated that the coronal suture is mesoderm-derived [32,33] while a more recent study showed that while the midsuture and parietal bone edge of the suture are mesoderm-derived, the frontal bone edge of the suture is of cranial neural crest origin [34]. Therefore, coronal suture fusion could occur via a neural crest cell autonomous or non-cell autonomous effect upon the deletion of *Alpl* in cranial neural crest cells via P0-Cre. Craniosynostosis was not found in any of the *Alpl*^fl/fl^; *P0-Cre^+^* mice by 35 days post birth, a time at which calvaria growth is essentially complete [35]. These data indicate that the deletion of *Alpl* in paraxial mesoderm and/or the deletion of *Alpl* in both cranial neural crest and paraxial mesoderm causes craniosynostosis in the *Alpl*^−/−^ mouse model of hypophosphatasia and likely in individuals affected by hypophosphatasia.

Despite the lack of craniosynostosis, the *Alpl*^fl/fl^; *P0-Cre^+^* mice did exhibit an abnormal skull shape that was similar (increase in skull height; shorter anterior–posterior) and different (no increase in skull width) compared to the skull shape abnormalities seen in global *Alpl*^−/−^ mice. Both the global and cranial neural crest specific ablations of *Alpl* caused decreased anterior cranial base bone lengths and cranial base fusions, which could account for the decrease in total anterior–posterior skull length and the compensating increase in skull height. It is also possible that the abnormal skull shape in *Alpl*-deficient mice occurs due to direct effects of TNAP on cranial bone growth. Nasal bones are shorter in both *Alpl*^fl/fl^; *P0-Cre^+^* and *Alpl*^−/−^ mice, and we previously reported diminished proliferation and a defect in the cell cycle progression in *Alpl* deficient primary calvarial cells in culture [36,37]. TNAP is expressed in cranial bone rudiments several days prior to mineralization, but its function there is not yet established [38,39,40]. Future studies involving a more comprehensive and longitudinal analysis of cranial osteoprogenitor proliferation and cranial bone growth in *Alpl*^fl/fl^; *P0-Cre^+^* and/or *Alpl*^−/−^ mice are needed to determine whether cranial and/or facial bone growth differences impact the skull shape of *Alpl* deficient mice.

While we previously found the dysregulated development of both the ISS and SOS in global *Alpl*^−/−^ mice [15], in *Alpl*^fl/fl^; *P0-Cre^+^* mice, only the ISS was abnormal (Figure 3A) because *Alpl* was ablated in the ISS but not in the SOS of these mice. Presphenoid and basisphenoid bones surrounding the ISS were shorter, and the premature fusion of the ISS was found 80% of *Alpl*^fl/fl^; *P0-Cre^+^* as compared to *Alpl*^fl/fl^ mice at 35 days post birth. The histomorphometry of H&E-stained 5-day-old ISS sections revealed significantly increased hypertrophic zone widths in the *Alpl*^fl/fl^; *P0-Cre^+^* mice, similar to that seen in global *Alpl*^−/−^ mice [15]. This finding indicates that *Alpl* deficiency in local growth plate prehypertrophic and hypertrophic chondrocytes likely causes chondrocyte maturation defects, leading to cranial base defects, because *Alpl* is normally expressed in these cells.

Here we used previously verified P0-Cre to ablate TNAP in neural crest cells [41]. From a developmental standpoint, it is important to note that use of P0-Cre to ablate TNAP in neural crest cells can result in different phenotypes than when Wnt1-Cre is used to target neural crest cells [42,43]. P0-Cre is expressed in a different subset of neural crest cells than Wnt1-Cre during embryogenesis [16,43]. Therefore, it is possible that cranial neural crest cells originating from different early embryonic positions, as depicted by differential P0 and Wnt1 expression in embryogenesis, would lead to a different pattern of case cranial base defects upon TNAP ablation that could potentially include the SOS. We deem this unlikely because global are normal at birth [9,44], and we found no differences in the cranial base of newborn *Alpl*^fl/fl^; *P0-Cre^+^* as compared to *Alpl*^fl/fl^ mice (Appendix A). This indicates that TNAP is not essential for cranial base development until after birth. While the SOS still contains cells derived from neural crest and mesoderm at birth, previous reports showed the rostral movement of the neural crest/mesoderm boundary during postnatal cranial base development, leading ultimately to an SOS composed ultimately of mesoderm-derived cells at day 10 postnatal when labeled using a Wnt1-Cre driver [5], and at day 7 post birth, when labeled using a Po-Cre driver [6]. Consistent with the above findings that the SOS develops into a tissue of entirely mesoderm origin after birth, our results showed thebcomplete elimination of *Alpl* in the prehypertrophic and hypertrophic zones of the ISS but not the SOS of 5-day-old *Alpl*^fl/fl^; *P0-Cre^+^* mice via TNAP enzyme activity staining (Figure 3A), indicating that no cells of the SOS expressed P0 and so were mesoderm-derived at this time point.

Hypertrophic zone chondrocyte apoptosis was decreased in *Alpl*^fl/fl^; *P0-Cre^+^* mice, accounting for the expanded hypertrophic zone width and were similar to that seen in global *Alpl*^fl/fl^ mice [15] and in mouse models of hypophosphatemic rickets [45]. Proliferation was increased in the resting and proliferation zones of *Alpl*^fl/fl^; *P0-Cre^+^* mice, yet these zones were not increased in size compared to control mice, which could be explained if the proliferating cells of *Alpl*^fl/fl^; *P0-Cre^+^* mice were also enhanced in differentiation. Staining for Col X was increased in the *Alpl*^fl/fl^; *P0-Cre^+^* as compared to *Alpl*^fl/fl^ mice, indicating that premature differentiation was occurring in addition to increased proliferation. Sox9 expression was up in both resting and hypertrophic zones. Sox9 is a master transcriptional regulator of chondrocytes that promotes chondrocyte proliferation and inhibits maturation in resting and proliferating zones, while promoting maturation and hypertrophy in prehypertrophic zones, via the differential expression of co-transcription factors Sox5, Sox6m Runx2 and Mef2c [46,47,48]. Sox9 could therefore be mediating the ISS chondrocyte changes seen downstream of *Alpl* ablation in cranial neural crest cells. Col2 expression was not altered in *Alpl*^fl/fl^; *P0-Cre^+^* ISS (Appendix A).

Indian Hedgehog (IHH) and Parathyroid Hormone Related Peptide (PTHrP) have long been known to play an essential role in chondrocyte proliferation and maturation in long bone growth plates [17,49,50]. The IHH-PTHrP feedback loop is also considered essential for cranial base growth plates [51,52]. PTHrP is secreted from resting zone chondrocytes to promote proliferation and inhibit chondrocyte maturation, while also inhibiting the production of IHH. More distant cells in the prehypertrophic and hypertrophic zones express IHH, which in turn promotes proliferation and differentiation, while also stimulating PTHrP expression to impact resting zone chondrocytes. Because this signaling feedback loop is considered essential for the regulation of growth plates and subsequent endochondral bone growth, we investigated the expression of IHH and PTHrP in the ISS of *Alpl*^fl/fl^; *P0-Cre^+^* mice as compared to control mice. We found the increased expression of IHH in prehypertrophic and hypertrophic zones, in addition to the increased expression of PTHrP in the hypertrophic and resting zones of *Alpl*^fl/fl^; *P0-Cre^+^* ISS. Because TNAP and IHH are normally expressed in prehypertrophic and hypertrophic zones, our data suggest that TNAP deficiency in these zones causes high IHH expression, in turn leading to the ectopic expression of PTHrP in the hypertrophic zone as well as increased PTHrP expression in the resting zone, where PTHrP is normally expressed.

The apoptosis of hypertrophic zone chondrocytes was previously shown to be stimulated by inorganic phosphate (P_i_) and inhibited by inorganic pyrophosphate (PP_i_) [22,45]. In *Alpl* deficient cells, P_i_ levels should decrease while PP_i_ levels increase. To distinguish between P_i_ and PP_i_ effects on ISS hypertrophic zone chondrocyte apoptosis, we next isolated the ISS from wild type mice and cultured them for five days in media +/− P_i_ and PP_i_. Treatment with P_i_ increased the resting zone and proliferating zone widths, which is not entirely surprising considering that P_i_ is a known mitogen [20]. We were surprised to find that culture with P_i_ increased ISS hypertrophic zone widths because, as stated above, P_i_ was previously shown to be essential for promoting hypertrophic chondrocyte apoptosis in mice. Treatment with PP_i_ increased the hypertrophic chondrocyte width and decreased apoptosis, demonstrating that high PP_i_ levels downstream of TNAP deficiency [10,37] potentially caused the increased ISS hypertrophic zone widths seen in the *Alpl*^fl/fl^; *P0-Cre^+^* mice.

We next examined the expression of IHH and PTHrP upon treatment +/− P_i_ and PP_i_. We found that P_i_ but not PP_i_ increased the expression of IHH. Treatment with P_i_ also increased the expression of PTHrp, but PP_i_ did so to a much greater extent. Because P_i_ levels are anticipated to be low in *Alpl* deficient cells and tissues, these data again support a hypothesis that high PP_i_ levels mediate some of the abnormal ISS development seen in *Alpl*^fl/fl^; *P0-Cre^+^* as compared to control mice. We also found that P_i_ increased while PP_i_ decreased Col X and Sox9 expression which is not consistent with the idea that decreased P_i_ or increased PP^i^ downstream of TNAP deficiency in *Alpl*^fl/fl^; *P0-Cre^+^* ISS mediated the increase Col X and Sox9 expression found in vivo. The differences seen between organ culture and in vivo immunofluorescent expression of IHH, PTHrP, Col X and Sox9 could be explained by the absence of factors that may mediate the downstream effects of P_i_ in organ culture that are present in vivo and/or due to the fact that we utilized newborn ISS for the organ culture studies. Importantly, we noted some de-differentiation of cells in the dissected and cultured cranial base (this is why we had to combine chondrocyte zones in the organ cultured ISS). Yet chondrocyte zones are clearly distinguishable in sections taken directly from newborn mice (Appendix A). Because the staining of ISS sections taken directly from 5-day-old mice showed different effects of *Alpl* deficiency on chondrocyte maturation, apoptosis and signaling that were chondrocyte zone-specific, we hypothesize that the differences between the in vitro and in vivo data are at least in part due to the fact that the chondrocytes in cranial base organ culture were de-differentiating. Regardless, the results show that inorganic phosphate and pyrophosphate stimulate changes in cranial base chondrocytes that include signaling.

To establish effects of *Alpl* deficiency on earlier lineage neural crest cells, we transduced the O9.1 neural crest cell line with *Alpl* or control nontarget shRNA. The O9-1 cell line stably expresses stem cell markers and neural crest markers and is capable of contributing to cranial neural crest fates [28,53]. We then assessed the expression of stem cell and neural crest markers in these cells. Minimal differences were seen in the known markers of neural crest between the cell genotypes, other than in Sox9. Sox9 is known to be essential in early neural crest cells, likely for maintaining pluripotency [54], and later in establishing a chondrogenic lineage [55]; in addition, it has a previously discussed ability to control the proliferation and maturation of growth plate chondrocytes [56]. Lower Sox9 expression in the O9.1 cells could be an artifact of in vitro experimentation or indicate that *Alpl* deficiency in pluripotent cranial neural crest cells has dual effects, one earlier in development that lowers the tendency for initial chondrocyte fate commitment and one later in development (studied here) for altering growth plate chondrocyte maturation. The suppression of *Alpl* by shRNA decreased the expression of stem cell markers Nestin and Sca1 but increased the expression of CD44. Future studies are needed to establish a role for *Alpl* (TNAP) in neural crest chondrocyte lineage specification.

Overall, our results validate that *Alpl* expression is required in cranial neural crest cells for normal anterior cranial base development but not for craniosynostosis. *Alpl* deficiency in the intersphenoid synchondrosis (ISS) leads to aberrant Sox9 and IHH-PTHrP signaling, with increased proliferation, abnormal chondrocyte maturation and diminished apoptosis (Figure 8). Cranial base organ culture experiments appear to demonstrate that the apoptotic abnormalities are also mediated at least in part by increased levels of PP_i_ that occur downstream of *Alpl* (TNAP) deficiency.

## 4. Materials and Methods

### 4.1. Animals

Mice carrying Cre recombinase driven by the protein zero (P0) promoter (*P0-Cre* mouse, *C57BL/6J-Tg*(*P0-Cre*)*94Imeg*) were generously provided by Yuji Mishina (University of Michigan, School of Dentistry, Ann Arbor, MI, USA). Floxed *Alpl* (*Alpl*^fl/+^) were generously provided by José Luis Millán (Sanford Children’s Health Research Center, Human Genetics Program, La Jolla, CA, USA). *Alpl*^fl/+^ and *P0-Cre* mice were bred onto a 97% 129/SvJ (Jackson Laboratory, Bar Harbor, ME, USA) and 3% C57BL/6J (Charles River Laboratory, Wilmington, MA, USA) genetic background for a minimum of 7 generations prior to breeding for experimentation. *P0-Cre* transgenic mice were bred with *Alpl*^fl/+^ mice to generate *Alpl*^fl/+^; *P0-Cre^+^* mice, which were then bred with *Alpl*^fl/fl^ mice to generate *Alpl*^fl/fl^; *P0-Cre*, *Alpl*^fl/fl^ and *Alpl*^fl/+^ mice. Litter-matched *Alpl*^fl/fl^; *P0-Cre^+^* (experimental mice) and *Alpl*^fl/fl^ (control mice) were used in this study. The genotyping of the mice was determined via the following specific primers using tail DNA. Primers A (5′-TTG CGA TGT GTG AAG ATG TCC-3′) and B (5′-GGC TTG CTG TCG CCA GTA AC-3′) were used to detect the flox allele (expected size: wild type band is 224 bp, floxed band is 258 bp). Primers C (5′-ATG GTG TTG CCG CGC CAT CTG CCA-3′) and D (5′- CTA ATC GCC ATC TTC CAG CAG GCG-3′) were used to detect the Cre-recombined allele (expected size: 298 bp). Animals were maintained and used in compliance with institutional animal care protocols of the University of Michigan’s University Committee on Use and Care of Animals, and in accordance with federal guidelines for and use and care of animals in research. Because all the transgenic animals remained healthy and viable, no animals were excluded from the study. The experimental unit for in vivo studies was one mouse. Sample sizes for postnatal day 5 analyses were *n* = 6. Sample sizes for postnatal day 35 analyses were *n* = 20. The larger sample size used in day 35 analyses was based upon prior data generated by our lab, which indicated need for a sample size this large to reach statistical conclusions regarding craniosynostosis and cranial base fusions due to phenotype severity variability seen in *Alpl*^−/−^ mice. Mouse genotype was blinded for analyses then re-identified for statistical comparisons. Primary endpoints were to confirm knockout of *Alpl* in cranial neural crest but not paraxial mesoderm-derived skull tissues (e.g., the ISS but not the SOS), as well as the incidence of craniosynostosis and the incidence of cranial base fusions. Secondary endpoints were analyses of cranial base zones, as well as chondrocyte signaling, proliferation, differentiation and apoptosis.

### 4.2. Nano Computed Tomography

Whole skulls of 35-day-old *Alpl*^fl/fl^; *P0-Cre^+^* and *Alpl*^fl/fl^ mice were collected, fixed in 4% paraformaldehyde, serially dehydrated and stored in 70% ethanol at 4 °C until scan. Whole skull specimens were scanned at 80 kV and 400 µA with a 0.381 mm aluminum filter in a Nanotom M scanner (Waygate Technologies LP, Pasadena, TX, USA). In total, 1500 projection images were acquired at a source-to-axis distance of 48 mm for a resolution of 12 µm/voxel. Three-dimensional reconstructed nano-CT images were analyzed using Dragonfly image analysis software (Version 2021.1.0.977; Object Research Systems, Inc, Montreal, QC, Canada). No significant difference between sexes was found; therefore, sexes were combined for analyses (*n* = 20 per *Alpl*^fl/fl^; *P0-Cre^+^* and *Alpl*^fl/fl^ mice).

### 4.3. Linear Skull and Cranial Base Measurements

Linear craniofacial measurements between landmarks were calculated using an electronic digital caliper (Automation and Metrology Inc., Mentor, OH, USA) on whole dissected skulls. Linear skull and cranial base measurements were performed using previously defined skull landmarks (Appendix A) [15,57,58]. Measurements were performed twice for each measurement with the average used as a given measurement per mouse. The measurements included standard craniofacial measurements used by the Craniofacial Mutant Mouse Resource of Jackson Laboratory (Bar Harbor, ME, USA), which are nasal bone length (landmark 1–2), nose length (landmark 1–3), inner canthal distance (landmark 8–9), skull width (landmark 5–6), skull length (landmark 1–7), frontal bone length (landmark 2–3) and parietal bone length (landmark 3–4). The Jackson Laboratory skull height measurement was substituted with a cranial height measurement taken between the pari (landmark 4) and the inferior portion of the spheno-occipital synchondrosis (landmark 10), due to the elimination of the mandible in our study.

### 4.4. Histology and Histomorphometry

Whole skulls were dissected from *Alpl*^fl/fl^; *P0-Cre^+^* and *Alpl*^fl/fl^ mice at postnatal days 0 and 5 days. The skulls were fixed in the 4% paraformaldehyde (PFA) for 24 h. The specimens were cryoprotected in 30% sucrose in 0.01 mol/L PBS, embedded in an embedding medium (O. C. T. compound, Fisher Healthcare, Waltham, MA, USA), frozen on dry ice, and kept at −80 °C until sectioning. Serial sagittal sections were cut at thickness of 14 µm and processed for H&E (hematoxylin and eosin) staining, Alcian Blue staining, alkaline phosphatase (ALP) staining or immunofluorescence. Zone area, zone fraction and cell number in synchondrosis were analyzed using Image J software (version 1.50i).

For Alcian Blue staining, the specimens were brought to room temperature (RT), then fixed with 4% PFA for 15 min, washed in PBS 5 min × 3 times, treated with 3% acetic acid for 5 min then Alcian Blue 8GX (Sigma-Aldrich, St. Louis, MO, USA) for 10 min and then counterstained with nuclear fast red. For ALP activity staining, the specimens were brought to room temperature, then washed in acetone for 15 min, TBST (Tris-buffered saline pH 8.0, 1% Tween-20) for 60 min, then NTMT (0.1 M NaCl, 0.1 M Tris-HCl pH 9.5, 50 mM MgCl_2_, 0.1% Tween-20) for 60 min at 4 °C and stained with nitro-blue tetrazolium chloride (NBT) and bromo-4-chloro-3′-indolyphosphate *p*-toluidine (BCIP) at room temperature for 10 min, then counterstained with nuclear fast red.

### 4.5. Immunofluoresence

Primary antibodies and their corresponding antigen retrieval methods are shown in Table 1. Sections were postfixed in 4% PFA for 10 min at room temperature. For immunostaining, sections were permeabilized with 0.3 M glycine/0.25% Triton X/PBS for 60 min, blocked with 5% skim milk/10% goat serum/0.05% Triton X/PBS for 60 min, incubated with each antibody overnight at 4℃, and subsequently treated with goat anti-rabbit IgG antibody-conjugated Alexa Fluor 555 (1:400, Invitrogen) with 4′6-diamidino-2-phenylindoledihydrochloride (DAPI; ProLong Gold antifade reagent, Invitrogen) to stain for the nucleus. Apoptotic cells were detected using a TUNEL-based in situ cell death detection kit (Invitrogen) according to the product’s procedure. We performed the quantification of TUNEL by counting all TUNEL+ cells relative to the total area of the anterior and/or posterior hypertrophic zones of the ISS or SOS. Immunofluorescence negative controls were performed by replacing the primary antibody with PBS: these showed no specific immunoreaction. Digital images were taken using a Nikon Eclipse E800 microscope with an attached CCD camera. To create dual-color images, the images of the same field obtained with fluorochrome were merged using image-processing software (Affinity Photo, version 2.2).

### 4.6. O9-1 Cell Culture and Procedures

O9-1 cells (neural crest cell line) originally obtained from MilliporeSigma were generously provided by Dr. Vesa Kaartinen (University of Michigan, School of Dentistry, Ann Arbor, MI, USA). These cells were cultured in basal media containing DMEM (Gibco), 15% fetal bovine serum (FBS), 0.1 mM non-essential amino acids (Gibco), 2 mM l-glutamine (Gibco), 55 µM beta-mercaptoethanol (Gibco), 1 mM sodium pyruvate (Gibco), 100 U/mL penicillin and 100 µg/mL streptomycin (Gibco) that had been conditioned by SNL feeder cells (ATCC, Manassas, VA, USA) overnight. The media were filtered (0.22 µm pore size) and supplemented with 25 ng/mL basic fibroblast growth factor (bFGF; R&D Systems, Minneapolis, MN, USA) and 1000 U/mL leukemia inhibitory factor (LIF; Millipore, Burlington, MA, USA).

O9-1 cells were expanded on a fibronectin-coated plate (obtained from Sigma-Aldrich). O9-1 cells were seeded at 10,000–15,000 cells/cm^2^, followed by 3 days of culture to reach confluence for freezing or plating for experimentation. To induce chondrocyte differentiation, the monolayer culture was initially cultured in early differentiation media (α-minimum essential medium (MEM), 10% FBS, 100 U/mL penicillin, 100 µg/mL streptomycin, 0.1 mM dexamethasone, 5 mM NaPO_4_, 50 µg/mL ascorbic acid and 100ng/mL BMP2 (R&D Systems) for 3 days. Then, cells were trypsinized and cultured in a micromass format in a chondrogenic medium (α-MEM, 5% FBS, 1% ITS; Sigma-Aldrich, Burlington, MA, USA), 100 U/mL penicillin, 100 µg/mL streptomycin, 10 ng/mL TGF-β3 (R&D Systems), 50 µg/mL ascorbic acid, 10 ng/mL BMP2, 0.1 µM dexamethasone and 1 mM sodium pyruvate for an additional 7 days.

Alkaline phosphatase enzyme activity was assayed using the colorimetric substrate, NBT/BCIP (Sigma-Aldrich). For Alcian Blue staining, cells were washed with phosphate-buffered saline, fixed with 10% formalin, then stained with a 1% Alcian Blue solution (1% Alcian Blue in 60% ethanol/40% acetic acid). After destaining in 60% ethanol/40% acetic acid, cells were photographed with microscope and each well was scanned. Scanned wells (*n* = 3 per genotype) were quantified via densitometry using Image J software.

### 4.7. RNA Analysis

For RNA analysis, RNA was isolated using Trizol reagent (Invitrogen, Carlsbad, CA) following manufacturer protocols. mRNA levels were assayed via reverse transcription and real-time PCR. Real-time PCR was performed utilizing the murine *Gapdh* primer/probe set Mm99999915_g1, the murine *Alpl* primer/probe set Mm00475834_m1, the murine *Sox9* primer/probe set Mm00448840_m1, the murine *AP-2α* primer/probe set Mm00495574_m1, the murine *Twist1* primer/probe set Mm00442036_m1, the murine *Snail1* primer/probe set Mm00441533_m1 (neural crest cell markers), the murine *Nestin* primer/probe set Mm00450205_m1, the murine primer/probe set *CD44* Mm01277163_m1, the murine *Sca1* primer/probe set Mm00618853_m1 (Stem cell markers) and Taqman Universal PCR Master Mix (Applied Biosystems, Waltham MA, USA). Real-time PCR was performed on a ViiA7 thermocycler (Life Technologies, Carlsbad, CA, USA) and quantified via comparison with a standard curve. Results are presented as normalized to *Gapdh* mRNA levels.

### 4.8. Organ Culture and Procedures

Cranial base structures were dissected at birth from mice and cultured in 12-well plates. Cultures were submerged in 1 mL Dulbecco’s modified Eagle’s medium (DMEM, Gibco, Billings, MT, USA), containing 100 µg/mL Ascorbic acid, 2 mg/mL insulin/transferrin/selenium (ITS, Sigma-Aldrich), 100 U/mL Penicillin G and 100 µg/mL streptomycin (Gibco) and amphotericin B (Gibco). Experimental cultures were exposed to sodium P_i_ at 1 mM, sodium PP_i_ at 0.5 mM, or without PP_i_/P_i_. The medium was changed every two days.

### 4.9. Statistical Analyses

All quantified data are presented as means +/− standard deviations. Statistical analyses were performed using GraphPad Prism version 9.5 (GraphPad Software Inc., San Diego, CA, USA). Student’s *t*-test was used for two-group comparisons and the ANOVA was used for multiple-group comparisons. Fisher’s exact test was used to analyze the incidence of ISS fusion in 35-day-old mice. A *p*-value of <0.05 was considered statistically significant.

## Figures and Tables

**Figure 1 ijms-24-15401-f001:**
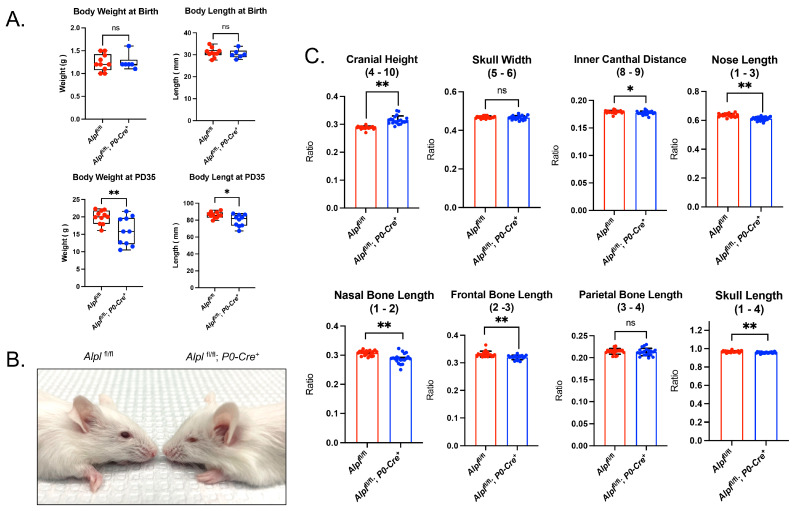
***Alpl* deletion by P0-Cre in cranial neural crest cells leads to skull shape defects.** (**A**) Body weight and body length in newborns were not different between genotypes. Body weight and body length were decreased in *Alpl^fl/fl^; P0-Cre^+^* as compared to *Alpl*^fl/fl^ mice by postnatal day 35 (P35). (**B**) Representative 35−day−old mice are shown. Note the midface hypoplasia evident in *Alpl* ^fl/fl^; *P0-Cre*^+^ but not in *Alpl*^fl/fl^ mice. (**C**) Linear skull measurements taken using digital calipers on dissected skulls, normalized to a total skull length measurement are shown. Length measurements were normalized to a total skull length measurement to take into account differences in skull size between genotypes. Size−normalized length measurements demonstrate craniofacial shape abnormalities in *Alpl*^fl/fl^; *P0-Cre*^+^ mice at P35. As compared to *Alpl*^fl/fl^ mice, *Alpl*^fl/fl^; *P0-Cre*^+^ mice exhibit significantly increased cranial height, with significantly diminished inner canthal distance, nose length, nasal bone length, frontal bone length and skull length. Skull width and parietal bone length are not different between genotypes. Overall, the skull shape of *Alpl*^fl/fl^; *P0-Cre*^+^ mice is proportionally shorter in anterior–posterior length and taller in height than the skull shape of *Alpl*^fl/fl^ mice. * *p*< 0.05, ** *p*< 0.01 vs. *Alpl*^fl/fl^, ns = nonsignificant. Red = *Alpl*^fl/fl^, Blue = *Alpl*^fl/fl^; *P0-Cre*^+^. *n* = 20 per genotype.

**Figure 2 ijms-24-15401-f002:**
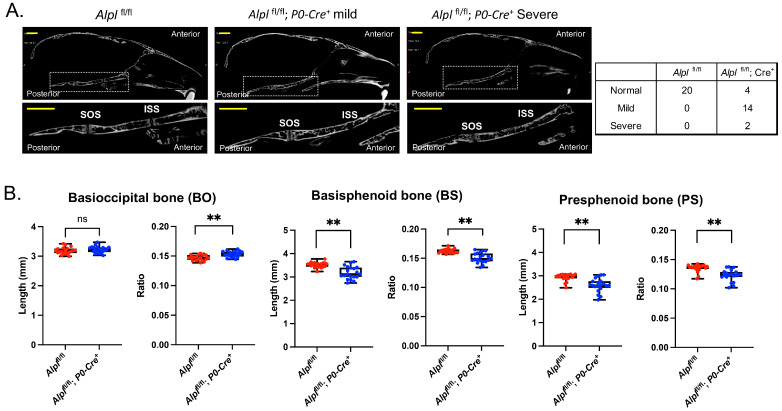
***Alpl* deletion by P0-Cre in cranial neural crest cells leads to cranial base defects.** (**A**) Nano-CT sagittal slice images of representative 35-day-old *Alpl*^fl/fl^ and *Alpl*^fl/fl^; *P0-Cre^+^* skulls are shown. The dotted box and its magnified image show the cranial base. SOS = spheno-occipital synchondrosis; ISS = intersphenoid synchondrosis. Close examination of cranial base nano-CT images of *Alpl*^fl/fl^, *P0-Cre^+^* mice showed partial to complete fusion of the ISS, which was not seen in any *Alpl*^fl/fl^ mice. The SOS appeared normal in all mice regardless of genotype. Because *Alpl*^fl/fl^; *P0-Cre^+^* present with varying phenotype severities, *Alpl*^fl/fl^; *P0-Cre^+^* mice of mild and severe phenotypes are shown. Numbers of *Alpl*^fl/fl^; *P0-Cre^+^* mice categorized by phenotype severity are also shown. (**B**) Nano-CT based measurements of cranial base bone lengths (mm), and the lengths after normalization by total skull length (ratio) between genotypes are shown. Cranial base measurements were normalized for total skull length to reveal relative differences because the skulls of *Alpl*^fl/fl^, *P0-Cre^+^* mice are overall smaller than that of their control littermates. Non-normalized basisphenoid and presphenoid bone lengths are diminished in *Alpl*^fl/fl^; *P0-Cre^+^* mice as compared to *Alpl*^fl/fl^ mice. After normalization for total skull size, the basioccipital bone length is increased while basisphenoid and presphenoid bone lengths are decreased in *Alpl*^fl/fl^; *P0-Cre^+^* mice as compared to *Alpl*^fl/fl^ mice, indicating. Yellow scale bars = 200 µm. ** *p*< 0.01 vs. *Alpl*^fl/fl^, ns = nonsignificant. Red = *Alpl*^fl/fl^, Blue = *Alpl*^fl/fl^; *P0-Cre^+^*. *n* = 20 per genotype.

**Figure 3 ijms-24-15401-f003:**
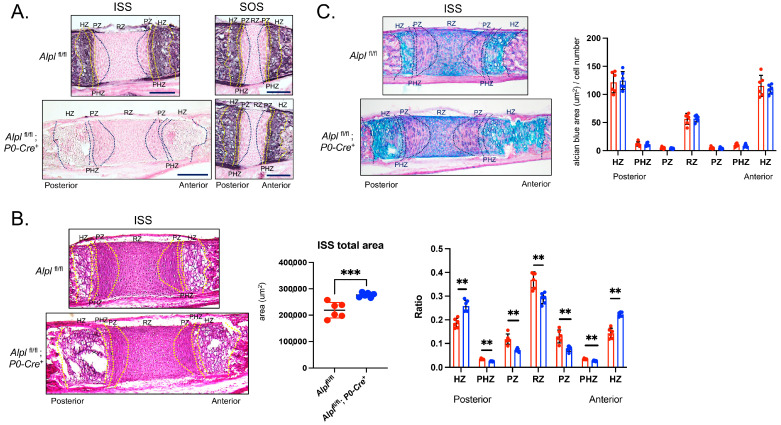
***Alpl* deletion by P0-Cre in cranial neural crest cells leads to hypertrophic zone expansion in the intersphenoidal synchondrosis at postnatal day 5.** (**A**) Staining for TNAP enzyme activity in the intersphenoid synchondrosis (ISS) and spheno-occipital synchondrosis (SOS) demonstrates a lack of TNAP in the perhypertrophic and hypertrophic zones and the surrounding bones of the ISS but not the SOS of *Alpl*^fl/flfl^; *P0-Cre*^+^ mice, which is indicative of cranial neural crest-derived specific *Alpl* ablation at this stage of development. (**B**) Hematoxylin and Eosin staining showing the histology of the ISS with/without the ablation of *Alpl* in cranial neural crest-derived cells is shown. The total area of the ISS is greater in *Alpl*^fl/fl^; *P0-Cre^+^* as compared to *Alpl*^fl/fl^ mice. Therefore, zone widths (ratio) are presented as normalized to total ISS length. Hypertrophic zone widths are significantly greater in *Alpl*^fl/fl^; *P0-Cre^+^* as compared to *Alpl*^fl/fl^ mice. All other zones are significantly decreased in *Alpl*^fl/fl^; *P0-Cre^+^* as compared to *Alpl*^fl/fl^ mice. (**C**) Alcian Blue staining for cartilaginous glycosaminoglycans in the ISS of *Alpl*^fl/fl^ and *Alpl*^fl/fl^; *P0-Cre^+^* mice. More Alcian Blue staining is evident in the ISS of *Alpl*^fl/flfl^; *P0-Cre*^+^ compared to *Alpl*^fl/fl^ mice, which is indicative of greater width of the hypertrophic zone but not when normalized by cell number. No differences in the Alcian Blue stain were observed when normalized by cell number, demonstrating similar amounts of glycosaminoglycan per cell. *n* = 6 per genotype. Black scale bars = 200 µm. ** *p*< 0.01, *** *p* < 0.005 between genotypes. Red = *Alpl*^fl/fl^, Blue = *Alpl*^fl/fl^; *P0-Cre^+^.* Zones were demarcated according to the following criteria. RZ: The nucleus is round. PZ: The nucleus is flat. PHZ: Adjacent to the PZ, and the cell shape is flattened and enlarged. HZ: The shape is greatly enlarged.

**Figure 4 ijms-24-15401-f004:**
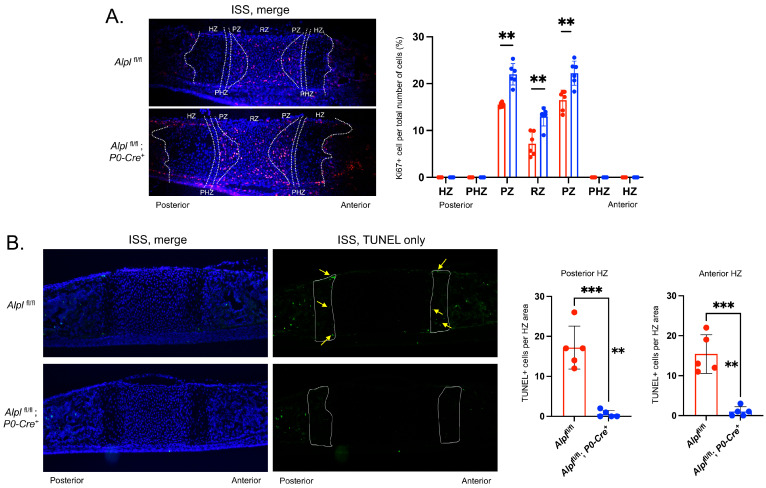
***Alpl* deletion by P0-Cre in cranial neural crest cells leads to increased proliferation and diminished apoptosis in the ISS at postnatal day 5.** (**A**) Cells were immunostained for Ki67 (red) as a marker of proliferating cells and co-stained with nuclear DAPI stain (blue). Merged stains are shown. Ki67+ cells relative to the total number of cells per chondrocyte zone were quantified to reveal increased proliferation in the proliferative and resting zones of the ISS. (**B**) Cells were immunostained with TUNEL (green) to detect apoptotic DNA breaks and co-stained with nuclear DAPI stain (blue). Merged and TUNEL-only stains are shown. TUNEL+ cells relative to the total area of the anterior and posterior hypertrophic zones (HZ) were quantified (HZ outlined in yellow, yellow arrows point to TUNEL+ cells) and show diminished apoptosis in the hypertrophic zones. *n* = 6 per genotype. ** *p* < 0.01, *** *p* < 0.005 between genotypes. Red = *Alpl*^fl/fl^, Blue = *Alpl*^fl/fl^: *P0-Cre^+^*.

**Figure 5 ijms-24-15401-f005:**
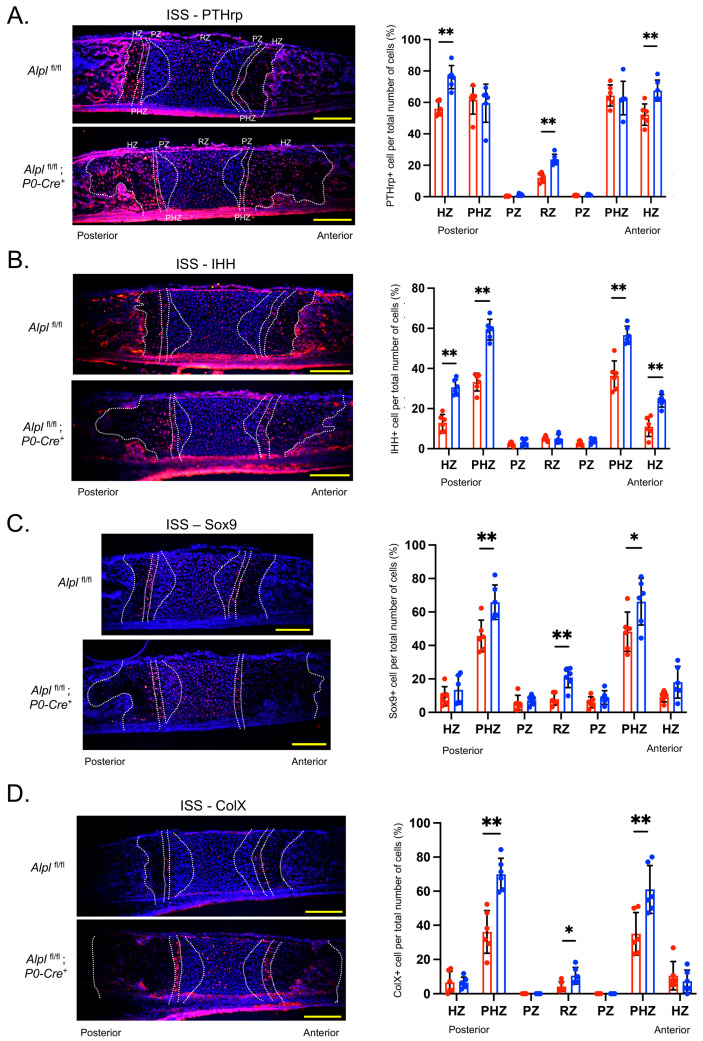
***Alpl* deletion by P0-Cre in cranial neural crest cells alters expression of PTHrP, IHH, Sox9 and ColX in the ISS at postnatal day 5.** (**A**) Cells were immunostained for PTHrp (red) and co-stained with nuclear DAPI stain (blue), then quantified by cell count as normalized to total cell number per chondrocyte zone. Quantification shows increased PTHrp expression in the hypertrophic and resting zones of *Alpl*^fl/fl^; *P0-Cre^+^* mice as compared to control mice. (**B**) Cells were immunostained for IHH (red) and co-stained with nuclear DAPI stain (blue), then quantified by cell count as normalized to total cell number per chondrocyte zone. Quantification shows increased IHH expression in the prehypertrophic and hypertrophic zones of *Alpl*^fl/fl^; *P0-Cre^+^* mice as compared to control mice. (**C**) Cells were immunostained for Sox9 (red) and co-stained with nuclear DAPI stain (blue), then quantified by cell count as normalized to total cell number per chondrocyte zone. Quantification shows increased Sox9 expression in the prehypertrophic and resting zones of *Alpl*^fl/fl^; *P0-Cre^+^* mice as compared to control mice. (**D**) Cells were immunostained for ColX (red) and costained with nuclear DAPI stain (blue), then quantified via cell count as normalized to total cell number per chondrocyte zone. Quantification shows increased ColX expression in the hypertrophic and resting zones of *Alpl*^fl/fl^; *P0-Cre^+^* mice as compared to control mice. *n* = 6 per genotype. Yellow scale bars = 200 µm. * *p* < 0.05, ** *p*< 0.01 between genotypes. Red = *Alpl*^fl/fl^, Blue = *Alpl*^fl/fl^; *P0-Cre^+^* mice.

**Figure 6 ijms-24-15401-f006:**
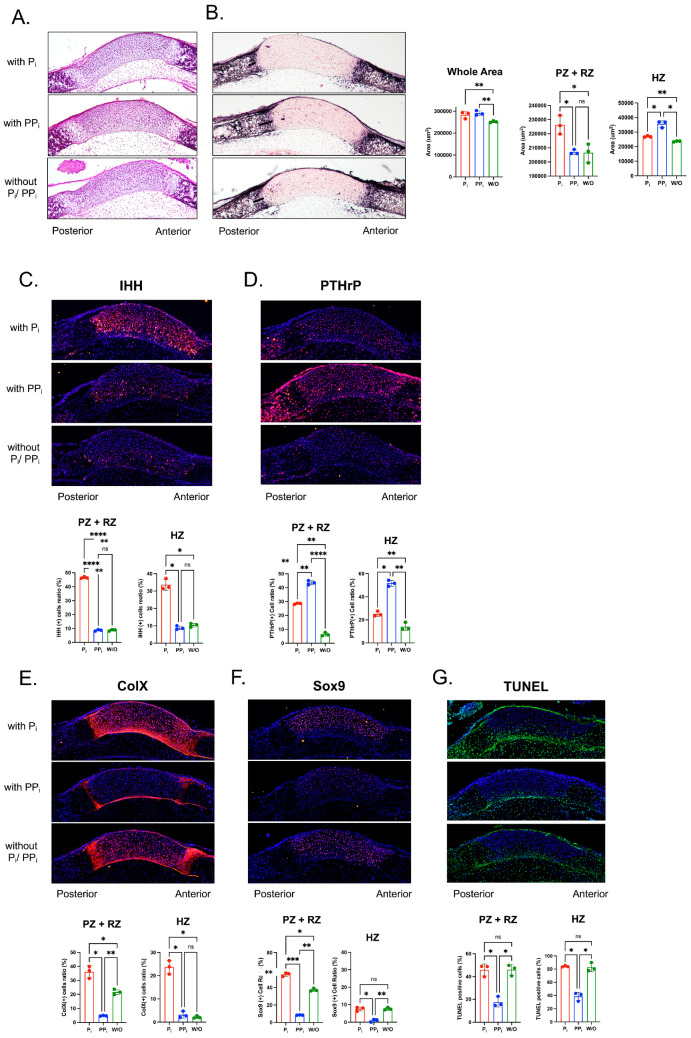
**Inorganic phosphate (P_i_) and inorganic pyrophosphate (PP_i_) differentially alter cranial base signaling and differentiation in culture.** Cranial base was dissected from newborn wild type mice and the ISS was cultured in P_i_ free media that was supplemented with/without P_i_ or PP_i_, as indicated. Only the posterior HZ was measured because no bone mineralization was apparent on the anterior aspect of the ISS at birth. Moreover, the RZ and PZ could not be distinguished. Thus, we measured the zone of PZ + RZ and the posterior HZ. (**A**) Hematoxylin and eosin (H&E) staining and (**B**) TNAP enzyme activity stains of cranial base sections after 5 days of culture with/without (P_i_) or (PP_i_). Quantification of growth plate zones demonstrates that culture with Pi or PPi increases overall area of the ISS, culture with Pi increases area of the resting (RZ) and proliferating zones (PZ), and culture with PPi increases area of the hypertrophic zone (HZ). There were no significant differences in TNAP enzyme activity between treatment groups. (**C**) P_i_ but not PP_i_ increases IHH signaling (red) in cultured ISS. DAPI nuclear stain is blue. (**D**) PP_i_ and, to a lesser extent, P_i_ increase PTHrp signaling (red) in cultured ISS. DAPI nuclear stain is blue (**E**) P_i_ increases while PP_i_ decreases ColX expression (red) in cultured ISS. DAPI nuclear stain is blue (**F**) P_i_ increases while PP_i_ decreases Sox9 expression (red) in cultured ISS treatment groups. DAPI nuclear stain is blue (**G**) PP_i_ decreases apoptosis as indicated by TUNEL staining (green) for DNA breaks. DAPI nuclear stain is blue * *p*< 0.05, ** *p*< 0.01, *** *p* < 0.005, **** *p* < 0.001, ns = nonsignificant, between indicated treatment groups.

**Figure 7 ijms-24-15401-f007:**
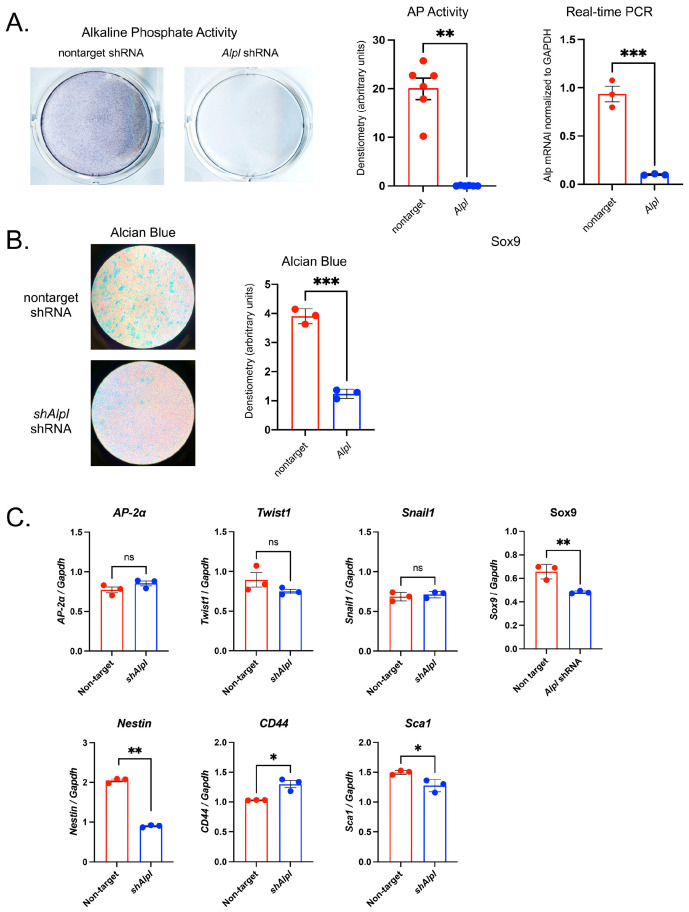
***Alpl*-deficient O9-1 neural crest cells exhibit diminished proteoglycan accumulation as well as the aberrant expression of neural crest and stem cell markers**. Cells were transduced with *Alpl* shRNA (sh*Alpl*) to knock down TNAP expression or control nontarget shRNA. TNAP enzyme activity was visualized through the incubation of cells with a colorimetric substate and quantified by densitometry. mRNA levels were quantified via real time PCR and are presented as normalized to GAPDH. (**A**) TNAP enzyme activity and *Alpl* mRNA levels are significantly reduced in *shAlpl* O9-1 cells as compared to the control cells. (**B**) Cells were cultured under chondrocyte differentiation conditions for 7 days, stained with Alcian Blue for cartilaginous glycosaminoglycans, then quantified by densitometry. Alcian Blue staining for proteoglycan accumulation was significantly diminished in *shAlpl* O9-1 cells as compared to the control cells. (**C**) The expression of Sox9 was diminished in the *Alpl* deficient as compared to control cells, while other markers of the neural crest were not different between genotypes. The expression of Nestin and Sca1 decreased while the expression of CD44 increased in *Alpl* deficient as compared to control cells. Red = non-target O9-1 cells; blue = *shAlpl* O9-1 cells. * *p*< 0.05, ** *p*< 0.01, *** *p* < 0.005 between genotypes.

**Figure 8 ijms-24-15401-f008:**
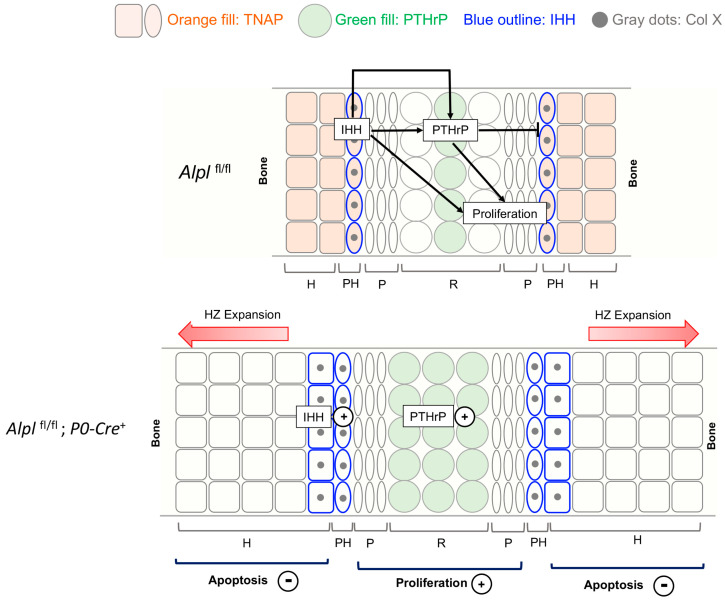
**Model depicting the major changes caused by conditional *Alpl* deletion in cranial base development.** The intersphenoidal synchondrosis (ISS) is a cranial base growth plate that grows in an analogous fashion to long bone growth plates but is bidirectional in anatomy and function. The IHH-PTHrP signaling loop is essential for control of chondrocyte proliferation and maturation. IHH is expressed primarily in prehypertrophic chondrocytes where PTHrP is low. IHH acts to promote chondrocyte proliferation and chondrocyte maturation, while also stimulating PTHrP expression. PTHrP acts on resting zone chondrocytes to maintain chondrocyte proliferation and inhibit differentiation/maturation. In control mice, TNAP is expressed in prehypertrophic zone and hypertrophic zone cells. Ablation of *Alpl* increases both IHH and PTHrP expression. Ablation of *Alpl* also increases proliferation, increases ColX expression and diminishes apoptosis leading to hypertrophic zone expansion. While not shown here, ablation of *Alpl* via cranial neural crest *P0-Cre* also increases expression of Sox9 in both resting and prehypertrophic chondrocytes. Sox9 is a master transcriptional regulator that promotes chondrocyte proliferation and inhibits maturation in resting and proliferating zones, while promoting maturation and hypertrophy in prehypertrophic zones, via differential expression of co-transcription factors. Because *Alpl* is co−expressed with Sox9 and with IHH in control mice, we hypothesize that *Alpl* ablation increases chondrocyte proliferation, increases ColX expression and diminishes apoptosis through increased Sox9 expression and/or increased IHH expression (IHH in turn would increase PTHrP expression). Increased levels of inorganic pyrophosphate (PP_i_) may also contribute to the diminished apoptosis and hypertrophic zone expansion in *Alpl* (TNAP) deficient mice, as this effect was strongly seen in the ISS organ culture studies.

**Table 1 ijms-24-15401-t001:** Primary antibodies.

Antibody	Company	Cat. #	Type	Anti-Retrieval	Dilution
Ki67	Abcam, Cambridge, UK	ab16667	Rabbit Monoclonal	0.5% Trypsin 37 °C 20 min	200
PTHrP	Invitrogen (Thermo Fisher Scientific, Waltham, MA, USA)	PA5-102455	Rabbit Polyclonal	No	500
IHH	Abcam	ab39634	Rabbit Polyclonal	No	100
Col 2	Abcam	ab34712	Rabbit Polyclonal	pH6.0 5.5 mM Citric Acid 90 °C 20 min, RT 20 min	100
Col X	Cosmo Bio USA, Carlsbad, CA, USA	LSL-LB-0092	Rabbit Polyclonal	No	100
Sox9	Abcam	ab185966	Rabbit Monoclonal	No	500

## Data Availability

The data presented in this study are available on request from the corresponding author.

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
