# Peer review of "Cranial Neural Crest Specific Deletion of Alpl (TNAP) via P0-Cre Causes Abnormal Chondrocyte Maturation and Deficient Cranial Base Growth"

_ijms, 2023, doi:10.3390/ijms242015401_

Round 1

Reviewer 1 Report

In this paper the authors present the craniofacial phenotype of mice in which they have deleted the Alpl gene, which encodes TNAP, using a the P0-Cre deletor mouse which targets predominantly Cranial Neural Crest Cells (CNCCs) and derived tissues. The article follows initial studies in which the authors have generated and analyzed mice carrying a general inactivation of the Alpl gene, these mice are considered models of the  human condition hypophosphatasia, which is associated to TNAP (Alpl) deficiency.

The paper is well written and well documented and deserves consideration for publication.

The main aim of the paper is to understand the contribution of CNCCs to the craniofacial phenotype of Alpl-/- mice. The article focuses primarily on two synchondroses of the cranial base : the  spheno-occipital synchondrosis (SOS) and the intersphenoid synchondrosis (ISS). Both SOS and ISS are affected in Alpl-/- mice, while, in this study, the authors show that ISS is the only affected after a P0-cre deletion.

The authors claim that while the ISS is CNCCs-derived, the SOS derives primarily from the paraxial mesoderm and this explains the difference observed, however, lineage studies (McBratney-Owen et al 2008) have shown that the developing SOS has a mixed CNCCs/Mesodermal  origin as seen in in X-Gal stained crania of Wnt1-Cre/R26R (Wnt1-cre is another CNCCs reporter mouse similar, but non identical to the P0-cre used in this study) and Mesp1-Cre/R26R mice.

My main question is to understand why the SOS and the ISS have a different phenotype in the Alplf/f/PO-cre mice and in the Alpl-/- mice. This difference might derive from a fine difference between of the pattern of cre expression of P0-cre and Wnt1-cre mice which has been used for CNCCs lineage (either Wnt1-cre would hit some of the mesoderm or the P0-cre has an incomplete targeting of CNCCs or a different spatio/temporal pattern of expression) or could derive from the fact that the CNCCs contribution is not particularly important for SOS development. As the SOS is located exactly at the CNCCs/mesodermal border, a precise answer to this question could be important for embryologists. A possible answer to this question could be obtained by generating a few  Wnt1-cre/Alplf/f and comparing their phenotype to that of P0-cre/Alplf/f mice. As Wnt1-cre does not hit the mesoderm if the phenotype obtained includes a defect of SOS one should think that the differences observed depends on a spatio/temporal restriction of P0 expression while, if the phenotype is identical (meaning if there is no effect of the Wnt1-cre deletion on the SOS) one could conclude that CNCCs do not have an important contribution to SOS differentiation. 

I encourage therefore the authors to perform these crossing as the results of these experiments, in my view, could permit to refine our knowledge of the CNCs/Mesodermal border and its post-natal evolution and will contribute valuable data to the debate on the use of Wnt1 or P0 for CNCCs deletion. These crossing will take some time and some effort, but, the results could be interesting in any case. 

Author Response

Response to Reviewer 1:

Thank you for reviewing our manuscript. In response to your highly appropriate query, while we have not performed Wnt1-Cre driven staining of cells in the SOS in mice, but we do now include an additional paragraph in the discussion section (lines 534-554) with additional references on this topic.

While the SOS is composed of mixed mesoderm/neural crest origin during embryogenesis and at birth, the mesoderm/neural crest boundary moves rostrally after birth such that the SOS is composed entirely of mesoderm derived cells several days after birth (McBratney-Owen et al., 2008; Wei et al., 2018). We agree that P0 and Wnt1 Cre are expressed in different subpopulations of neural crest cells during embryogenesis and that this has been shown in other studies to lead to different phenotypes when using them as drivers to conditionally knockdown gene expression (Kulkarni et al., 2018; Ueharu et al., 2022). Yet global Alpl-/- and P0-Cre conditional Alpl-/- mice are normal at birth with phenotypes developing postnatally. Our results showed complete elimination of Alpl via staining for TNAP enzyme activity in the prehypertrophic and hypertrophic zones of the ISS but not the SOS of 5-day-old Alplfl/fl;P0-Cre+ mice (Figure 3A), indicating that no cells of the SOS on either the anterior or posterior ends of this tissue had P0 mediated Alpl gene deletion, and so were likely mesoderm derived at this time point.

Additionally, the fact that we deleted Alpl via P0-Cre is now directly specified in the title and multiple figure legends in the manuscript.

Reviewer 2 Report

The article by Ohkura et al. entitled "Cranial neural crest specific deletion of TNAP causes abnormal chondrocyte maturation and deficient cranial base growth” explored the role of Alpl in mice cranial base development using tissue-specific cre, ex-vivo organ culture, and in-vitro cell culture. Overall, the article is clearly written, and the conclusions are supported by evidence.  The article will be much better if they work on the following.

1.     Instead of TNAP it is better to use Alpl in the mouse.

2.     Explain where the P0-cre is normally expressed.

3.     Provide details of the figures. For example, it is hard to understand what the numbers represent in Figure 1C.

4.     Start the sentence with words rather than numbers. (Eighty percent (80%) instead of 80%).

5.     Lebel the figure diligently. In Figure 3 legend, B is described as ALP staining, but it is clearly Hematoxylin and Eosin staining. C is labeled as Hematoxylin and Eosin staining but it is not.

6.     Provide high-resolution images of all the fluorescence images. At the current resolution it is very hard to see the positive signal, especially the green one. However, the article would benefit from having high-resolution images of all fluorescent images.

7.      Why do controls of ex-vivo specimens (Fig. 6C-G) show different staining patterns than controls from in vivo specimens (Fig. 5A-D). The control should show similar or near-similar staining patterns.  

Author Response

Response to Reviewer 2:

  1. Instead of TNAP it is better to use Alpl in the mouse.

TNAP will be replaced with Alpl many places within the title, abstract and manuscript text. In some places, Alpl (TNAP) will be used to minimize confusion to readers and clearly convey that TNAP is expressed downstream of Alpl in mice.

  1. Explain where the P0-cre is normally expressed.

In response to this request, and in addition to a query by Reviewer 2, we will include a paragraph describing expression of P0 and Wnt1 in neural crest cells, and use of these drivers to target gene expression in neural crest cells (discussion section, lines 534-554).

  1. Provide details of the figures. For example, it is hard to understand what the numbers represent in Figure 1C.

Thank you for this important critique. Detail in the legend of Figure 1C will be increased (new text in red) to ease interpretation of the figure. In addition, all other figure legends will be revised for improved clarity (some text was deleted, new text in red).

  1. Start the sentence with words rather than numbers. (Eighty percent (80%) instead of 80%).

This change will be included in the revised manuscript (line 138, red text).

  1. Label the figure diligently. In Figure 3 legend, B is described as ALP staining, but it is clearly Hematoxylin and Eosin staining. C is labeled as Hematoxylin and Eosin staining but it is not.

We again thank this reviewer for catching these figure and figure legend errors. We did find errors in multiple figure legends. All figures and figure legends will be revised for improved accuracy and clarity in this revised manuscript.

  1. Provide high-resolution images of all the fluorescence images. At the current resolution it is very hard to see the positive signal, especially the green one. However, the article would benefit from having high-resolution images of all fluorescent images.

We apologize for this. We do have high resolution images. These were downsized in creation of the PDF for manuscript upload. We provide the high resolution images with the uploaded PDF in a zip file, that we anticipate will be incorporated into the final manuscript before publication. Please let us know if you cannot access the higher resolution images.

  1. Why do controls of ex-vivo specimens (Fig. 6C-G) show different staining patterns than controls from in vivo specimens (Fig. 5A-D). The control should show similar or near-similar staining patterns.  

This is an important aspect of results that requires better discussion and interpretation. As written in the initial version of the manuscript, the cultured cranial base lacked surrounding tissues which may be important for control of chondrocyte signaling and maturation. Included in this revised manuscript is an additional point for interpretation: chondrocytes in culture appeared somewhat de-differentiated (we could not distinguish and therefore had to combine zones for the cultured ISS), but ISS chondrocyte zones are clearly distinguishable in sections taken directly from newborn mice (Supplementary Figure 3). Text describing this is now included in the discussion section (lines 612-621).

Round 2

Reviewer 1 Report

The authors have replied to my suggestions and, in my view, the paper can now be published.

Reviewer 2 Report

-Most of my concerns are addressed.